# Independent component analysis recovers consistent regulatory signals from disparate datasets

**Anand V. Sastry**[1], **Alyssa Hu**[1], **David Heckmann**[1¤], **Saugat Poudel**[1], **Erol Kavvas**[1], **Bernhard O. Palsson**[1,2]*

**1** Department of Bioengineering, University of California San Diego, La Jolla, California, United States of America, **2** Novo Nordisk Foundation Center for Biosustainability, Technical University of Denmark, Lyngby, Denmark

¤ Current address: Institute for Computer Science and Department of Biology, Heinrich Heine University, Düsseldorf, Germany

* palsson@ucsd.edu

**Data Availability Statement:** The authors confirm that all data underlying the findings are fully available without restriction. All code and data required to recreate the results and figures in this

## Abstract

The availability of bacterial transcriptomes has dramatically increased in recent years. This data deluge could result in detailed inference of underlying regulatory networks, but the diversity of experimental platforms and protocols introduces critical biases that could hinder scalable analysis of existing data. Here, we show that the underlying structure of the *E. coli* transcriptome, as determined by Independent Component Analysis (ICA), is conserved across multiple independent datasets, including both RNA-seq and microarray datasets. We subsequently combined five transcriptomics datasets into a large compendium containing over 800 expression profiles and discovered that its underlying ICA-based structure was still comparable to that of the individual datasets. With this understanding, we expanded our analysis to over 3,000 *E. coli* expression profiles and predicted three high-impact regulons that respond to oxidative stress, anaerobiosis, and antibiotic treatment. ICA thus enables deep analysis of disparate data to uncover new insights that were not visible in the individual datasets.

## Author summary

Cells adapt to diverse environments by regulating gene expression. Genome-wide measurements of gene expression levels have exponentially increased in recent years, but successful integration and analysis of these datasets are limited. Recently, we showed that independent component analysis (ICA), a signal deconvolution algorithm, can separate a large bacterial gene expression dataset into groups of co-regulated genes. This previous study focused on data generated by a standardized pipeline and did not address whether ICA extracts the same quantitative co-expression signals across expression profiling platforms. In this study, we show that ICA finds similar co-regulation patterns underlying multiple gene expression datasets and can be used as a tool to integrate and interpret diverse datasets. Using a dataset containing over 3,000 expression profiles, we predicted

manuscript are available at https://github.com/
SBRG/xplatform_ica_paper. Additional code is
available upon request.

**Funding:** AVS, AH, DH, SP, EK, and BOP were
funded by the Novo Nordisk Foundation Center for
Biosustainability and the Technical University of
Denmark (grant number NNF10CC1016517). The
funders had no role in study design, data collection
and analysis, decision to publish, or preparation of
the manuscript.

**Competing interests:** The authors have declared
that no competing interests exist.

three new regulons and characterized their activities. Since large, standardized expression
datasets only exist for a few bacterial strains, these results broaden the possible applica-
tions of this tool to better understand transcriptional regulation across a wide range of
microbes.

## Introduction

Publicly available datasets, such as the NCBI Gene Expression Omnibus (GEO) [1] and Array
Express [2], contain thousands of transcriptomics datasets that are often designed and ana-
lyzed for a specific study. Historically, microarrays were the platform of choice for transcrip-
tomic interrogation, resulting in large, publicly available datasets containing thousands of
expression profiles for a variety of organisms [3,4]. Over the past decade, usage of RNA
sequencing (RNA-seq) has surpassed microarrays due to its higher sensitivity and ability to
detect new transcripts [5].

Multiple consortia have performed extensive comparisons of expression levels across differ-
ent microarray and RNA-seq platforms [6–8]. These studies showed that absolute gene expres-
sion levels cannot be accurately measured by either expression profiling technique, whereas
relative abundances are consistent across a wide range of transcriptomics platforms with
appropriate quality controls. To further complicate matters, batch effects and technical hetero-
geneity continue to present significant challenges to successful integration of omics datasets
[9].

Differential expression analysis is the most common analytical method applied to transcrip-
tomics datasets. However, differential expression analysis is limited in dimensionality,
interpretability, and reproducibility; it can only be applied to pairs of experimental conditions,
requires additional analysis to interpret large swaths of differentially expressed genes [10,11],
and is highly dependent on the quantification pipeline [12,13]. Alternatively, machine learning
methods, especially matrix factorization [14,15], have provided new tools for extracting low-
dimensional biological information from large omics data.

In particular, independent component analysis (ICA) has been shown to extract biologically
significant gene sets from many transcriptomics datasets [16–22]. ICA outperformed 42 mod-
ule detection methods in a comprehensive examination across 5 organisms [23]. Previously,
we applied ICA to a high-quality *Escherichia coli* gene expression dataset generated from a
standardized protocol to extract 92 independently-modulated groups of genes (called iModu-
lons) [24]. Sixty-one of these 92 iModulons represented the targets of specific transcriptional
regulators and described their activities across every condition in the dataset. An additional 25
iModulons were linked to biological functions or genetic perturbations, leaving only 6 unchar-
acterized iModulons. iModulons have provided clear physiological explanations for transcrip-
tional changes in significantly perturbed cells [25–27] and were used to characterize a novel
adenosine transporter in *E. coli* [28].

We have also computed iModulons for *S. aureus* and *B. subtilis* using single-source expres-
sion datasets [29,30]. In addition, ICA has been applied to human transcriptomic datasets to
identify co-regulated gene sets [18,21], but characterization of many components was hindered
due to the high fraction of unknown human genes relative to model bacteria [31,32]. A recent
study applied multiple matrix factorization methods to 14 independent cancer expression
datasets that were generated using the same microarray platform. The study found that ICA
extracted many components that were conserved across multiple expression datasets, whereas
PCA and NMF did not identify reproducible components [33].

However, it is still unclear whether ICA extracts reproducible regulatory signals from expression compendia compiled from diverse sources, such as between unrelated RNA-seq and microarray datasets. In this study, we show that consistent regulatory components can be identified in expression datasets spanning disparate experimental conditions, and that these components are robust to dataset integration. Through analysis of five independent *E. coli* transcriptomics datasets, we identify a coherent structure without requiring batch normalization procedures. In addition, integrated analysis of the different datasets demonstrated compelling evidence towards regulon discovery. These results present ICA as a promising tool to integrate and understand the flood of omics data challenging scientists today.

## Results

### ICA identifies biologically-relevant "iModulons" in five transcriptomic datasets

We compiled two RNA-seq and three microarray datasets, each using a different expression profiling technology or generated from a different research group (Table 1 and S1–S3 Datasets). Each dataset was independently processed, centered, and decomposed with ICA (See Methods). This process generated a set of independent components (ICs) for each dataset that represent underlying signals (i.e., transcriptional regulators) in the transcriptional dataset (Fig 1A and S4 and S5 Datasets). Each IC contains a weighting for every gene; a high IC gene weighting indicates that gene expression is strongly influenced by the underlying signal or regulator, whereas a low weighting indicates that gene expression is unaffected by the regulator. As most transcriptional regulators only control a small set of genes, the majority of gene weightings were near zero for a given IC.

To understand the biological role of each IC, we selected genes in each IC with significant non-zero weightings and referred to this set of genes as an "iModulon" [24]. ICs, and their corresponding iModulons, represent the underlying structure of the transcriptome under any experimental condition in the database. The condition-dependent dynamics of gene expression are captured by the activities of the ICs (also referred to as iModulon activities, S6 Dataset). In this study, we focused on the condition-invariant structure of the transcriptome (i.e., ICs and their resultant iModulons), to determine if transcriptome structure is conserved across platforms.

We categorized the resulting iModulons from each dataset into four classes, based on their gene content: (1) Regulatory, (2) Functional, (3) Genomic, and (4) Uncharacterized (Fig 1B and 1C and S7 Dataset). Prior work showed that iModulons are highly consistent with, but not always identical to, known regulons. On average, these iModulons captured 80% of the known targets of their linked transcriptional regulators and have accurately predicted new regulon members [24]. Although iModulons often contain genes known to be regulated by a single transcriptional regulator (e.g., transcription factor, sigma factor, or riboswitch), it has been observed that iModulons may represent combinations of multiple regulons [18,24].

**Table 1. Summary of transcriptomic datasets.**

| Dataset | Platform(s) | Source(s) | # Samples | # Unique Conditions | # Total iModulons |
|---------|-------------|-----------|-----------|---------------------|-------------------|
| RNAseq-1 (PRECISE) | Illumina MiSeq, HiSeq, NextSeq & GAIIX | [24] | 278 | 163 | 91 |
| RNAseq-2 | Applied Biosystems 5500XL Genetic Analyzer | [34] | 84 | 28 | 52 |
| MA-1 | Affymetrix *E. coli* Antisense Genome Array | [35] | 260 | 115 | 103 |
| MA-2 | Affymetrix *E. coli* Antisense Genome Array | [36–38] | 124 | 39 | 58 |
| MA-3 | Affymetrix *E. coli* Genome 2.0 Array | [36,39–43] | 56 | 20 | 32 |

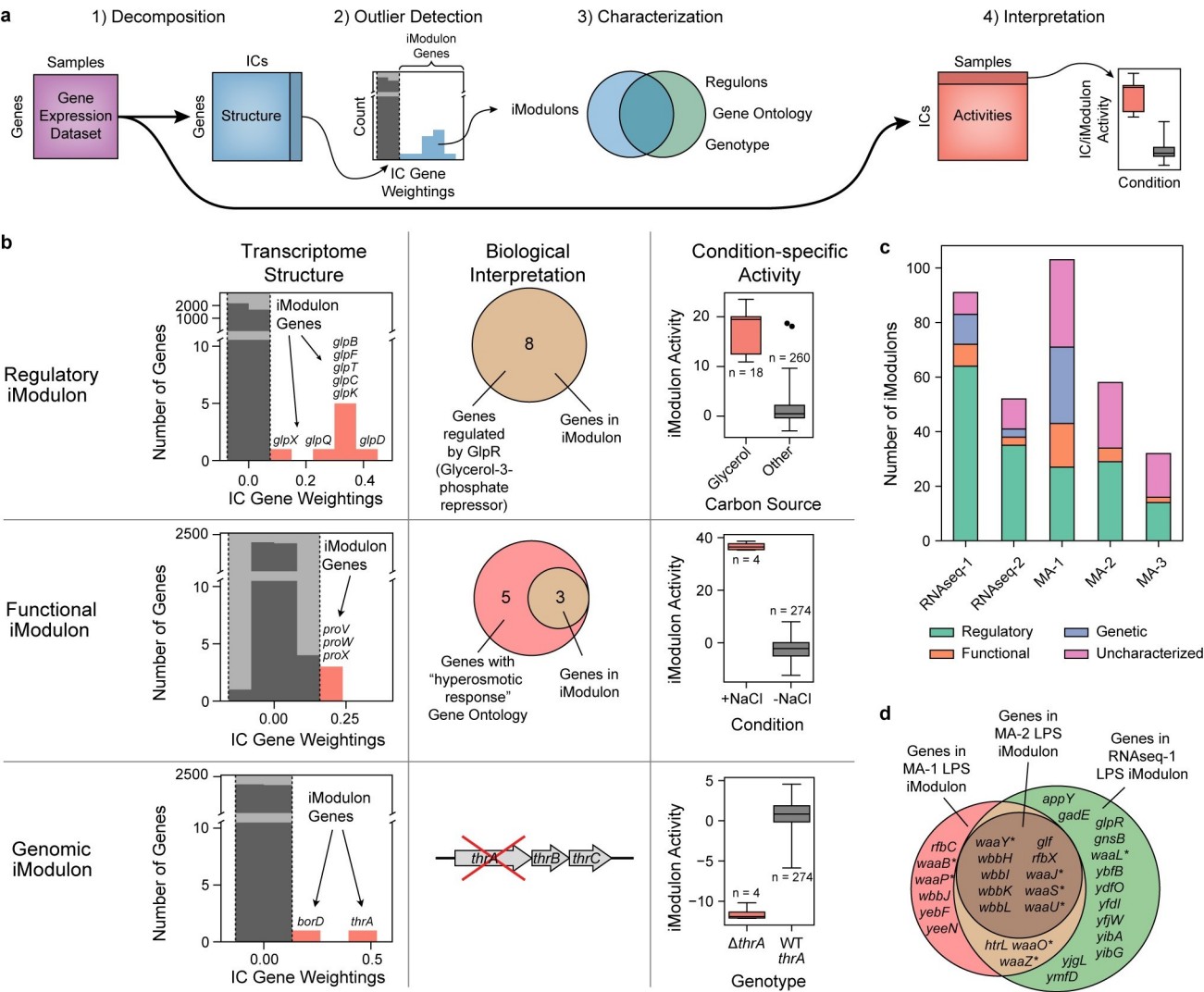

**Fig 1. Characterization of iModulons derived from five independent gene expression datasets.** (A) Schematic illustration of the workflow applied to each data set. (B) Descriptions of the three classes of characterized iModulons. The first column contains histograms illustrating the distribution of gene weightings in each of three independent components (ICs) from the RNAseq-1 dataset. Genes outside of a threshold (in red) belong to an "iModulon". The second column illustrates the biological interpretation of the iModulon types. iModulons are characterized by comparing their genes with known regulons, ontological annotations, and genotypes. The third column displays the ICA-computed activity levels for the selected ICs across all 278 conditions in the RNAseq-1 dataset. (C) Bar chart describing the four types of iModulons computed from each of the five gene expression datasets. (D) Comparison of three Functional iModulons, each derived from decomposition of a different dataset. Asterisks indicate genes annotated with the GO term "lipopolysaccharide core region biosynthetic process".

In this study, a *Regulatory* iModulon was defined as an iModulon that was statistically enriched with a single regulon (Fisher's exact test, FDR $< 10^{-5}$). Regulatory iModulons were named after the single transcriptional regulator whose targets provided the best overlap (See Methods).

*Functional* iModulons contained genes with highly similar functions but lacked common regulator(s). For example, three iModulons were identified in MA-1, MA-2, and RNAseq-1, respectively, that share 10 genes and were enriched in the gene ontology (GO) term "lipopolysaccharide core region biosynthetic process" (Fig 1D). These iModulons were named based on the GO term with the lowest enrichment p-value (FDR < .01). This category also included

iModulons composed of prophages with no known regulators. Functional iModulons represent a compelling opportunity for discovery of new transcriptional regulators [44].

*Genomic* iModulons reflect alterations in the genome, such as those resulting from engineered overexpression or knock-out of one or more genes. The activities of these iModulons represent the presence of these genomic alterations, such as large-scale duplications or deletions. These iModulons can be useful to validate strain-specific mutations or deletions [24] and successful plasmid transformations [45]. Genomic iModulons were categorized by reconciling iModulon genes and activities with strain-specific genotypes.

The remaining *Uncharacterized* iModulons could not be interpreted, either due to a high number of uncharacterized genes or the presence of seemingly functionally unrelated genes. Uncharacterized iModulons may represent undiscovered regulons, noise, or other unwanted sources of variation in the datasets.

The RNA-seq datasets produced the highest fraction of characterized iModulons; only 9% of iModulons in RNAseq-1, and 21% of iModulons in RNAseq-2 were Uncharacterized. In contrast, we were unable to characterize 30–50% of the iModulons derived from the microarray compendia. Uncharacterized iModulons exhibit significantly higher variance between replicates than other iModulons (Student's t-test p-value $< 10^{-10}$, S1 Fig), which indicates that the microarray datasets contain more noise-capturing uncharacterized iModulons than the RNA-seq datasets.

In summary, application of ICA to expression datasets identifies biologically-relevant iModulons in both microarray and RNA-seq datasets. These iModulons can be categorized as Regulatory, Functional, Genomic, or Uncharacterized. Uncharacterized microarray iModulons are more likely to represent technical noise in the dataset.

## An iModulon controlled by CysB exists in all five datasets

Each of the five transcriptomic datasets were created using different technologies or generated by different research groups, spanning fifteen years (2004–2019). In addition, each dataset contained vastly different experimental conditions and genotypes, such as overexpressed cellular division proteins in the MA-1 dataset [35], diverse nutritional supplements in the RNAseq-1 dataset, or strains evolved to resist antibiotics in the RNAseq-2 dataset [34]. Although the presence of different conditions in each dataset complicated the identification of consistent iModulons, many of the iModulons generated from the five datasets unexpectedly shared similar genes and annotations.

For example, each dataset produced an iModulon enriched in the sulfur utilization regulator CysB. On average, each CysB iModulon shared 87% of its genes with the other CysB iModulons. The IC gene weightings between four of these iModulons were also highly correlated (Pearson R > 0.5), indicating that the genes in the iModulons were modulated at similar ratios across all five datasets regardless of expression profiling platform (Fig 2A). However, the CysB iModulon from the MA-3 dataset was less correlated with the other iModulons. The MA-3 CysB iModulon contained many genes that were absent from any other CysB iModulon. Many of these genes were involved in the biosynthesis of other amino acids and were present in different iModulons in the other datasets. In fact, the MA-3 CysB iModulon could be approximated ($R^2$ = 0.47) by a linear combination of 10 iModulons from the RNAseq-1 dataset related to nutrient availability (Fig 2B and S1 Table). A similar analysis revealed at least three additional iModulons from the MA-3 dataset that were composed of a linear combination of RNAseq-1 iModulons (S2 Fig). The MA-3 dataset contained the least number of samples and resulted in the fewest number of iModulons.

The detailed investigation into the CysB iModulons revealed that iModulons can be found in multiple datasets containing a set of nearly identical genes. In addition, the IC gene

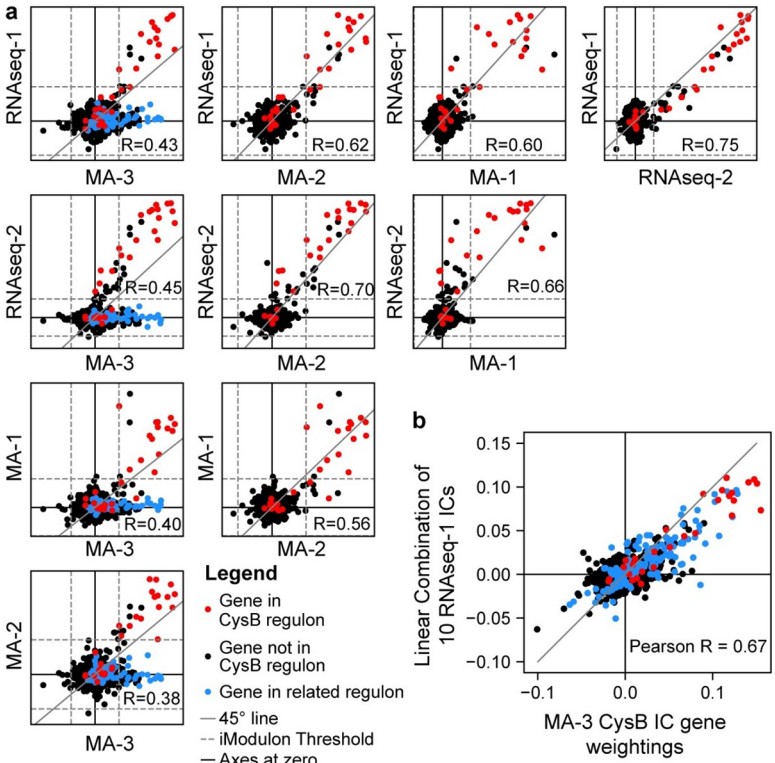

**Fig 2. Comparison of IC gene weightings for iModulons enriched with genes in the CysB regulon.** (A) Scatter plot between IC gene weightings for CysB-linked iModulons across all five datasets. Genes in the published CysB regulon are colored in red. For comparisons in the first column involving the MA-3 dataset, genes regulated by any of the following regulators are colored in blue: MetJ, TrpR, GlpR, ArgR, Lrp, CysB, leu-tRNA-mediated transcriptional attenuation, or thiamine riboswitch. All other genes are black. Dashed lines indicate iModulon thresholds. Gray solid lines indicate the 45-degree line of equal gene weightings. (B) Scatter plot between IC gene weightings for the CysB iModulon in MA-3 compared to a linear combination of 10 ICs from RNAseq-1 (see S1 Table). Color scheme is identical to panel (A).

weightings for overlapping iModulons are highly correlated and often nearly equal. Finally, we showed that iModulons generated from a small dataset (e.g., MA-3) are often linear combinations of iModulons from larger datasets. This observation demonstrates how ICA may be unable to differentiate all the underlying source signals from smaller datasets.

## The iModulon structure is conserved across transcriptomic datasets

To extend our assessment of iModulons reproducibility, we compared all iModulons found in each dataset using a Reciprocal Best Hit (RBH) graph [33] (Fig 3A and S2 Table). In the RBH graph, each node represents an iModulon, and nodes are connected when iModulons from two different datasets find each other as the best scoring iModulon in the other dataset [46]. iModulons were scored by the absolute correlation between the gene weightings in their respective ICs, as shown for the CysB iModulons in the previous section. Since RBHs are not always highly correlated, we trimmed the graph to exclude RBHs with a Pearson R < 0.3 (S3 Fig).

Of the 336 iModulons identified across all five datasets, nearly half (45%) were linked to an iModulon in another dataset. Of these 151 reproducible iModulons, 110 (73%) were classified as Regulatory, and the remaining were either Functional or Uncharacterized (Fig 3B). Only one Genomic iModulon was matched to an iModulon from another dataset (S4 Fig).

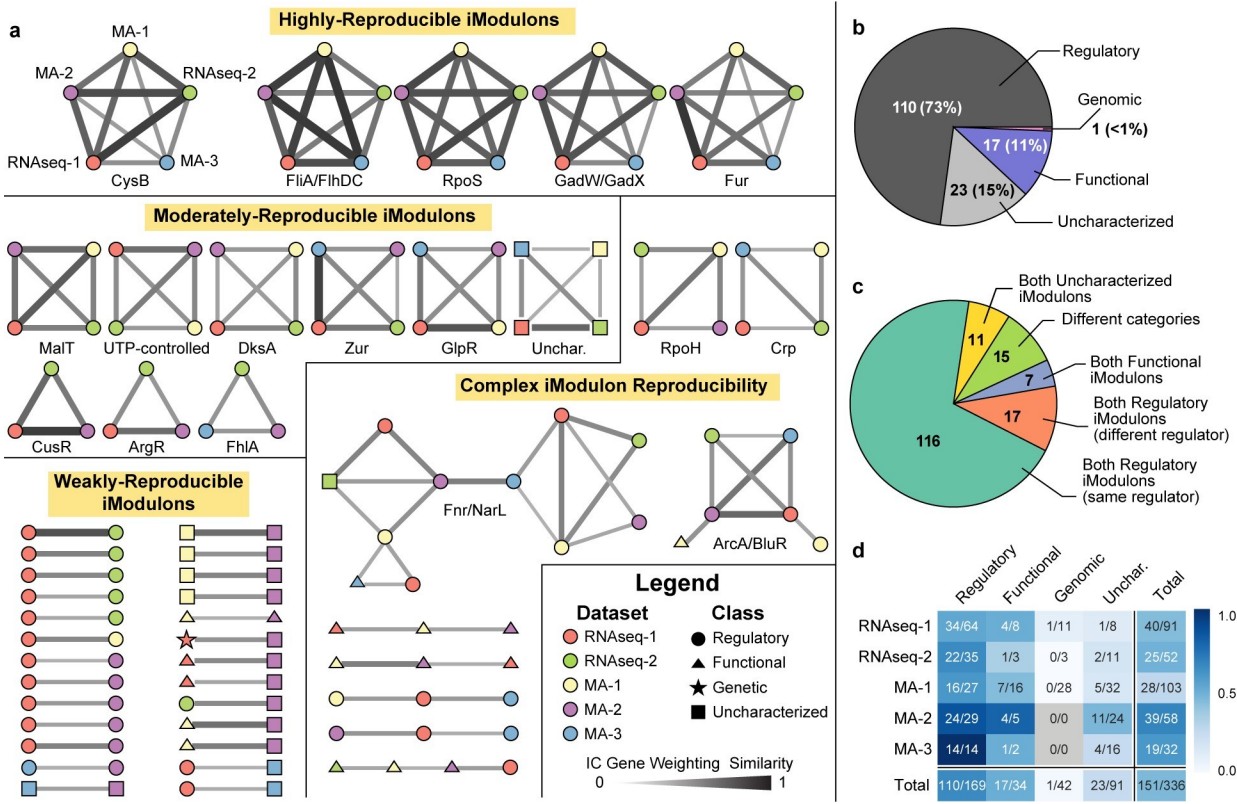

**Fig 3. The iModulon structure is conserved across five datasets.** (A) Reciprocal best hit (RBH) graph indicating iModulons as nodes and RBHs as edges. Node color indicates the source dataset for the iModulon, as denoted for the CysB iModulon. Node shape indicates the iModulon category. Edge thickness and darkness indicate the gene weighting similarity. Clusters are labelled with the regulator(s) that are linked to the iModulons in the cluster, if available, and grouped by level of iModulon reproducibility. (B) Pie chart describing the categories of all iModulons shown in the RBH graph. (C) Pie chart describing the types of edges in the RBH graph. (D) Heatmap indicating how many iModulons from each dataset and category were in the RBH graph.

Most edges in Fig 3A connected two Regulatory iModulons enriched in the same regulator (Fig 3C). Across all five datasets, Regulatory iModulons were the most reproducible category, whereas Genomic iModulons were the least reproducible (Fig 3D). The lowest fraction of reproducible Regulatory iModulons was observed in the RNAseq-1 dataset, likely due to niche transcription factors that are active under specific environmental conditions.

Many iModulons in the RBH graph were separated into well-defined clusters, each linking at most one iModulon from each dataset. iModulons in a cluster were often linked to the same regulator, as labelled below each cluster. Five clusters were highly-reproducible, each containing one iModulon from each of the five datasets. Three of these clusters were enriched with genes from a single regulon, whereas the other two clusters contained iModulons enriched with genes regulated by closely-related transcriptional regulators.

An additional nine iModulons were found in three or four datasets, forming moderately-reproducible clusters. The missing iModulons could either be explained by low variation of iModulon genes in the missing dataset, or by many-to-one linkages between iModulons that could not be captured by the RBH method, as demonstrated with the MA-3 iModulon (Figs 2B and S2).

A few clusters exhibited complex connectivity, the largest of which contained iModulons linked to either of the nitrate response regulators Fnr and NarL. Cellular respiration is regulated by a combination of highly-interconnected transcription factors [36], indicating that

these iModulons represent combined effects of regulators (S5 Fig). These examples highlight a limitation of the RBH method, as it is unable to detect whether an iModulon from one dataset is a linear combination of iModulons from another dataset.

The analysis presented in this section shows that ICA recovers many consistent, technology-independent signals across multiple gene expression datasets. This property is unique to ICA as other dimensionality reduction methods, such as principal component analysis, do not identify consistent components across datasets [33] (S6 Fig).

## iModulon structure is consistent across genetically diverse strains

Even though all five expression datasets have been previously analyzed and published, iModulons can uncover hidden information that was obscured in traditional differential expression analysis. Here, we will demonstrate how iModulons enable deep understanding of mutations acquired in laboratory evolutions.

The RNAseq-2 dataset contains 24 strains of *E. coli* that were evolved in the laboratory to tolerate 12 different antibiotics [47]. The evolved strains were re-sequenced [48], and expression profiled under identical environmental backgrounds [34]. Using the condition-specific iModulon activities, we could clearly connect mutations to distinct changes in gene expression.

For example, the MarA/Rob iModulon was highly active in all strains with a mutation in *marR*, *rob*, or *acrR* (Fig 4A). MarA and Rob are repressed by AcrR and MarR, indicating that the mutations disrupted these transcription factors to de-repress MarA and Rob. On the other hand, the Rob R156H mutation appears to increase the effectiveness of the activator. MarR, MarA, Rob, AcrR, SoxR, and SoxS are all involved in a complex network regulating genes related to antibiotic resistance and oxidative stress response (Fig 4B).

The SoxS iModulon also shows a clear pattern of activation that is correlated with the presence of *soxR* mutations (Fig 4C). SoxR activates SoxS, which upregulates genes related to oxidative stress. Even though the mutations in SoxR seem disruptive, including a truncation (L139*), the mutations appear to constitutively activate the protein. Similar mutations have been shown to constitutively activate SoxR and confer multiple antibiotic resistance in clinical isolates [49]. Although MarA, Rob, and SoxS share over 50% of their known gene targets, we found distinct iModulons for SoxS and MarA/Rob that shared 10 genes (Fig 4D). The iModulon activities for the SoxS and MarA/Rob iModulons show the strongest responses to mutations in SoxR and MarR, respectively.

Larger genomic rearrangements can also affect the structure of the transcriptome. As described previously, large deletions or duplications can result in Genomic iModulons that are active for single strains. In this dataset, strains exhibiting high RcsAB iModulon activity also harbored insertion sequence (IS)-mediated inversions that decrease the expression of the Lon protease [50] (Fig 4E and 4F). Although Lon degrades RcsA, SoxS, and MarA, the effects were clearest in the RcsAB iModulon. iModulons are a powerful tool to analyze new transcriptomic data or extract new knowledge from previously published datasets. iModulons are especially useful in connecting mutations to clear transcriptomic shifts, bridging the gap between genotype and phenotype.

## Data integration increases the resolution of iModulons

iModulons represent the underlying structure of transcriptomics datasets. Since elements of this structure were highly consistent across the five transcriptomics datasets, we combined the datasets together to explore whether ICA would identify a similar structure in the combined compendium, which contained 802 expression profiles (Fig 5A).

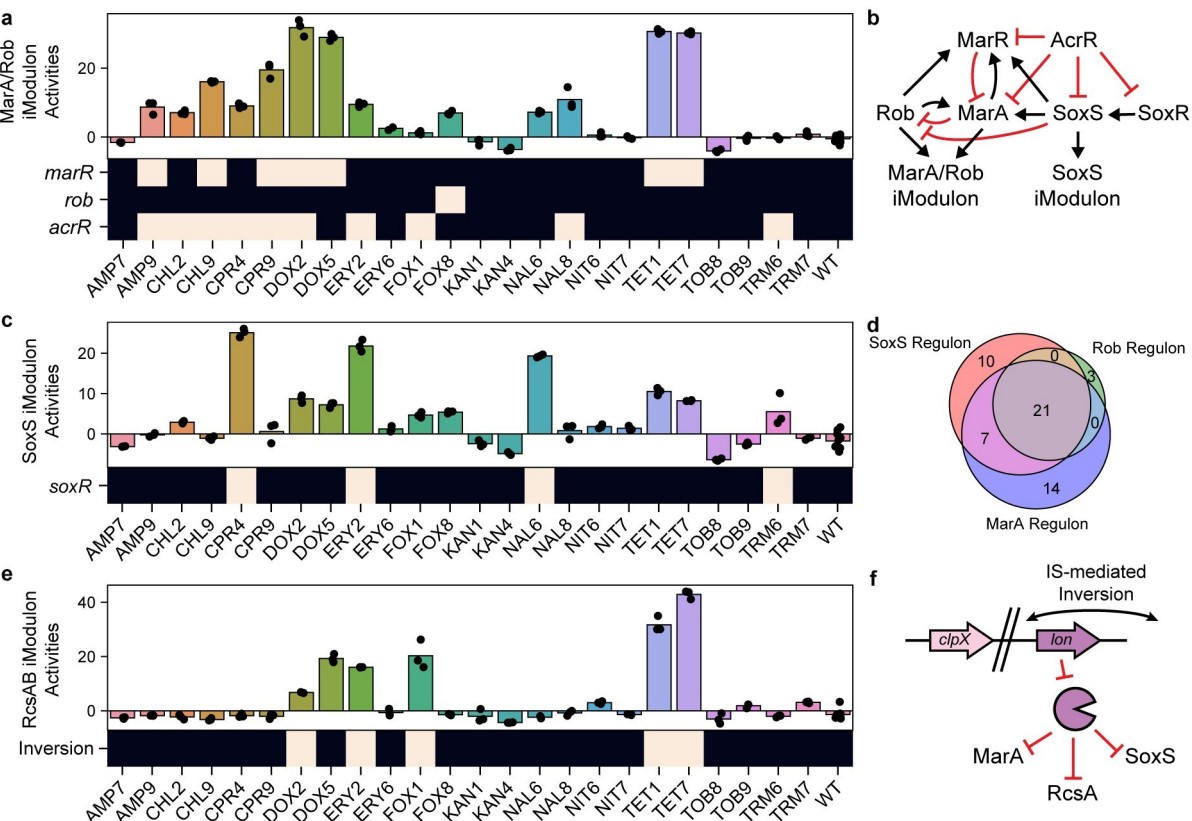

**Fig 4. iModulon activities capture the effects of mutations from adaptive evolution to multiple antibiotics.** (A) Bar chart of activities for the MarA/Rob iModulon. Individual points represent biological replicates. The heatmap below the bar chart shows the presence of mutations in the specified gene for the specified strain. Strain names are described in Lazar et al [34]. (B) Regulatory network for antibiotic resistance in *E. coli*. Black arrows indicate activation, and red arrows represent repression. Auto-regulation is not shown. (C) Bar chart of activities for the SoxS iModulon, similar to panel (A). Heatmap shows the presence of mutations in *soxR*. (D) Venn diagram showing the overlap of genes in the MarA/Rob iModulon and the SoxS iModulon. (E) Bar chart of activities for the RcsAB iModulon, similar to panel (A). Heatmap shows the presence of a genomic inversion. (F) Schematic illustration of the genomic inversion upstream of *lon*. The inversion decreases *lon* expression, resulting in longer residency times for MarA, SoxS, and RcsA.

We extracted 181 iModulons from the combined compendium and characterized them as described previously (Fig 5B). The majority of iModulons from this combined compendium (60%) were enriched with targets of a known transcriptional regulator. To understand the effects of dataset integration, we compared the new iModulons to the iModulons from the independent datasets.

First, we asked whether the iModulons identified in the original datasets were retained upon data integration. From the RBH graph, 75% of the 181 iModulons in the combined compendium could be directly linked back to at least one iModulon derived from an individual dataset, many of which were linked to iModulons from two or more datasets (Fig 5C and S3 Table). The two datasets with the largest number of unique conditions, (RNAseq-1 and MA-1) contained the most uniquely mapped iModulons, as these datasets were more likely to activate niche transcriptional regulators.

We next asked why some iModulons were missing after data integration. Nearly all of the 92 missing iModulons were categorized as either Genomic or Uncharacterized (Fig 5D). Missing iModulons were weaker signals, each of which accounted for a significantly lower fraction of expression variance than retained iModulons (Mann-Whitney-U Test p-value $< 10^{-5}$, S7A Fig).

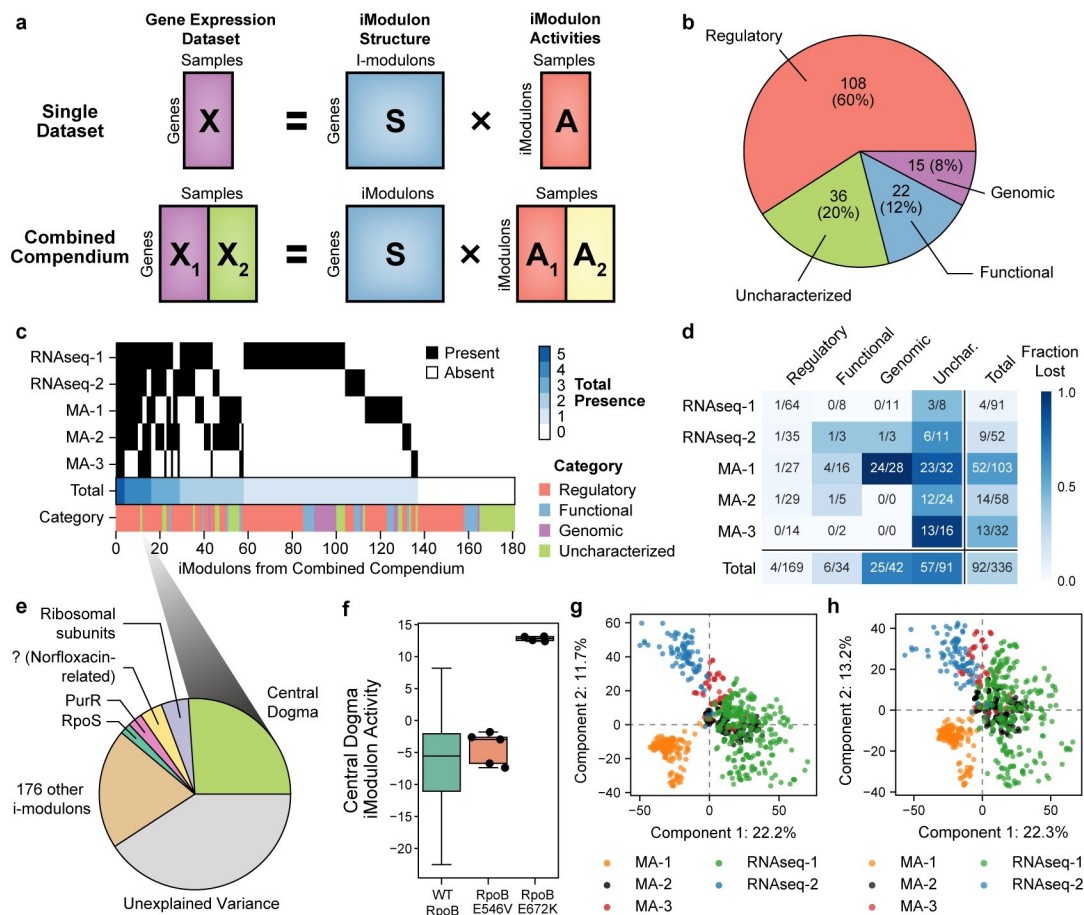

**Fig 5. Decomposition of combined compendium results in increased resolution** (A) Schematic illustration of data integration. (B) Pie chart showing the categories of the 181 iModulons from the combined dataset. (C) Heatmap illustrating which iModulons from the combined compendium were matched to iModulons in individual datasets in the RBH graph. The total number of matches for each iModulon is shown in blue. Category of each iModulon is shown below. (D) Heatmap illustrating how many iModulons from each category and dataset were not matched to an iModulon in the full compendium in the RBH graph. (E) Pie chart illustrating the fraction of expression difference resulting from ppGpp-RNAP binding explained by each iModulon. The norfloxacin-related iModulon is described in S7E Fig. (F) Boxplot of the Central Dogma iModulon activities for expression profiles in the RNAseq-1 dataset. (G) Principal component loadings of the combined compendium expression levels without batch correction. (H) Principal component loadings of the iModulon activities from the combined compendium.

On the other hand, we found 44 new iModulons in the combined compendium that could not be traced back to the iModulons from the individual datasets. Of these new 44 iModulons, 21 represented the effects of transcriptional regulators that could not be discriminated in any of the individual datasets alone, and 12 were dominated by a single gene (S7B and S7C Fig). A previous study showed that over-decomposition of a transcriptomic dataset results in smaller groups of genes found in each iModulon but does not affect the biological relevance of other iModulons [51].

Next, we investigated whether dataset integration could change the quality of the Regulatory iModulons. Since a Regulatory iModulon consists of genes in a known regulon, the quality of a Regulatory iModulon can be assessed using the F1-score, which is the harmonic mean of precision and recall of the iModulon compared to the regulon (S7D Fig). We inspected all

Regulatory iModulons that were found in both an individual dataset and the combined compendium and found that the average F1-score increased from 0.53 to 0.58 (Wilcoxon Rank Sum p-value = $10^{-4}$) (S7E Fig). This result showed that on average, Regulatory iModulons from the combined compendium were more similar to defined regulons than the Regulatory iModulons from individual datasets.

In summary, we found that many iModulons computed from a large multi-source dataset reflected the iModulons identified in the individual datasets. In addition, data integration led to (a) more iModulons that represent transcriptional regulation, and (b) higher quality iModulons that reflect known regulation.

## An iModulon is responsive to ppGpp

Eleven Uncharacterized iModulons in the combined compendium were connected to at least two iModulons in the individual decompositions, and likely represent true biological signals. For example, one Uncharacterized iModulon in the combined compendium was linked back to four of the five individual datasets. These four iModulons created the only uncharacterized moderately-reproducible cluster in Fig 3A. Although this iModulon was not enriched in any GO term, it contained many genes encoding functions related to the central dogma of molecular biology, such as rRNA and tRNA modification, RNases, and helicases (S7F Fig). Therefore, we named it the "Central Dogma" iModulon. Over half of the genes in this iModulon are not known to be regulated by any transcription factor, and the remaining genes are not enriched in any common regulator.

A recent study found that binding of ppGpp, the stringent response alarmone [52], to RNA polymerase (RNAP) directly down-regulated 428 genes [53], including 48 of the 54 genes in Central Dogma iModulon. The Central Dogma iModulon explained 26% of the expression variation between strains affected by ppGpp-RNAP binding and the wild-type strain (Figs 5E and S7G). Additionally, an RNAP mutant strain exhibits the highest Central Dogma activity in the RNAseq-1 dataset (Fig 5F), providing evidence that the point mutation affects ppGpp binding to RNAP [54]. We therefore propose that this iModulon represents the direct transcriptomic effect of ppGpp-RNAP binding.

This example shows that conserved uncharacterized iModulons may represent new transcription factors or other less-characterized forms of transcriptional regulation.

## iModulon activities cannot be compared across disparate datasets

Finally, we checked whether the iModulon activities were affected by dataset integration. When we apply ICA to an expression profiling dataset, we obtain the **M** matrix, encoding the iModulon structure, and an **A** matrix, which contains activity levels for each iModulon across every expression profile (Figs 1A and 5A). We have focused our analysis thus far on the invariant properties of the **M** matrix, and briefly discuss the effects of data integration on the **A** matrix. iModulon activities reflect the overall change in expression of the iModulon genes and serve as a proxy for transcription factor activities [26]. Therefore, it is important to ensure that the relative iModulon activities are unchanged upon dataset integration. The absolute Pearson R correlation weighting between the iModulon activities of linked components showed that most iModulon activities are unaffected by dataset integration (S7H Fig).

Previously, principal component analysis (PCA) of integrated expression datasets revealed that the source of each dataset was the dominant discriminator [55,56]. This result is recapitulated in our combined compendium (Fig 5G). However, since the iModulon structure of each dataset (encoded in **M**) is consistent across datasets, we see that technical heterogeneity is

stored in the activity matrix (**A**) (Fig 5H). Therefore, iModulon activities cannot be compared across datasets, but can still reliably be compared within the same dataset.

## Predicting new regulons using big data

Since ICA could accurately identify iModulons from a combined dataset containing multiple expression profiling platforms, we applied our iModulon analytical pipeline to the COLOMBOS microarray dataset containing over 3,000 *E. coli* expression profiles [3]. The COLOMBOS dataset reports all microarray datasets centered with respect to dataset-specific reference conditions, as we have done in the RNA-seq and microarray datasets discussed thus far.

We identified 243 iModulons in the COLOMBOS dataset, of which 119 were enriched with a transcriptional regulator (Fig 6A). Due to the large number of datasets, we did not annotate Genomic iModulons, resulting in 103 Uncharacterized iModulons. Using the RBH approach, over half of the iModulons extracted from the COLOMBOS dataset were linked to iModulons found in the combined compendium described in the previous section (Fig 6B). Most of the Regulatory iModulons shared between the compendia displayed near-perfect overlap (Fig 6C). An additional eight of shared iModulons were uncharacterized in both datasets, indicating that they may represent true biological signals rather than expression noise. We propose biological or regulatory roles for three of these iModulons.

The first Uncharacterized iModulon contained 4 genes in the COLOMBOS iModulon, and 6 genes in the combined compendium (Fig 6D). Five of these genes (*hiuH*, *msrP*, *msrQ*, *hprR*, *hprS*) are in close genomic proximity (Fig 6E) and have an upstream binding site for the two-component system response regulator HprR [57]. Additionally, expression of *msrP* and *msrQ* was found to be dependent on a functional HprRS system [58]. Although only *hiuH* has been validated as a member of the HprR regulon [57], we propose that this iModulon is regulated by HprR. iModulon activities provide additional support for this claim. The HprRS two-component system responds to hydrogen peroxide [57]. COLOMBOS contains a dataset treated with hydrogen peroxide, which exhibits high HprR iModulon activity (Fig 6F).

The second Uncharacterized iModulon contains 11 shared genes in four operons: *ynjXY-ZABCD*, *yedEF*, *yjiLM*, and *ynjE* (Fig 6G). Of these eleven genes, the only characterized gene is *ynjE*, a molybdopterin sulfurtransferase. Two additional genes have putative functions: *yedE* is a putative selenium transporter and *yedF* is a putative sulfurtransferase. This iModulon is more active in anaerobic conditions than aerobic conditions (Fig 6H), but the iModulon's activity is lower in Fnr and ArcA mutant strains than the wild-type strain under nitrate respiration (Fig 6I). Anaerobic and nitrate respiration require the use of molybdoenzymes, such as nitrate reductase and selenocysteine-containing formate dehydrogenases[59]. Altogether, we propose that this iModulon is related to molybdenum and selenium usage and is likely integral to the proper functioning of molybdoenzymes under fermentative conditions. Although it is not clear which regulator controls this iModulon, the regulator is likely activated by Fnr and potentially repressed by ArcA, based on the activity levels of the mutant strains.

The final Uncharacterized iModulon contains 15 shared genes, many of which are located on the inner membrane or in the periplasm (Fig 6J). Notably, the iModulon contains a periplasmic chaperone (*ivy*), an outer membrane metalloprotease (*loiP*), and a heat shock-induced lipoprotein (*hslJ*). Many genes in this iModulon have been associated with the envelope stress Rcs-phosphorelay [60]. This iModulon is specifically active when treated with cell envelope damaging antimicrobial agents, such as polymyxin B, colicin, and a combination therapy of cefsulodin and mecillinam (Fig 6K, 6L, and 6M). Polymyxin B destabilizes the cell membrane and is often used as a last-resort antibiotic for multi-drug resistant E. coli infections [61]. Even

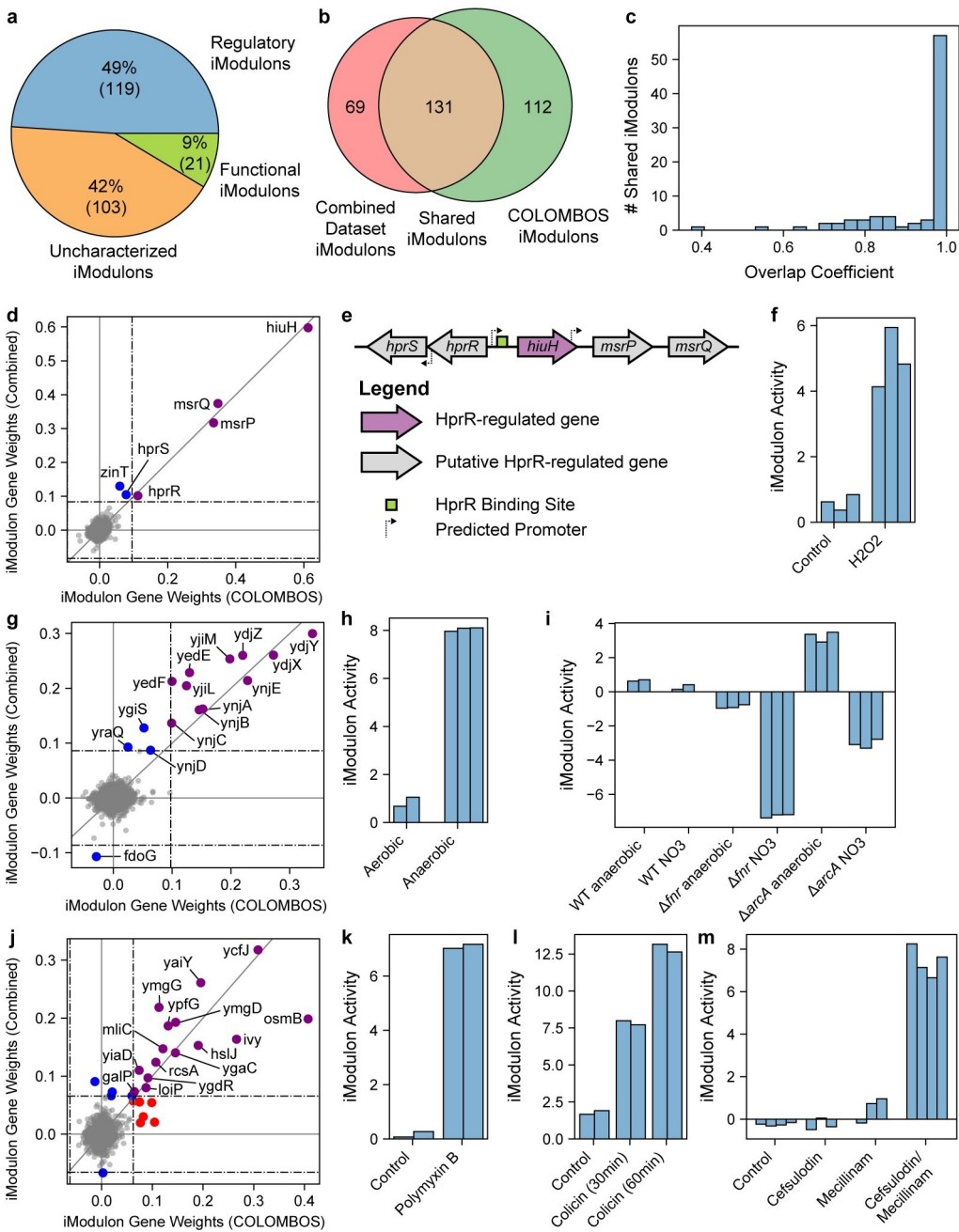

**Fig 6. Predicting regulons using big data.** (A) Pie chart displaying the number of Regulatory, Functional and Uncharacterized iModulons extracted from the COLOMBOS *E. coli* compendium. (B) Venn diagram illustrating the number of iModulons shared between the COLOMBOS compendium and the combined dataset discussed in Fig 5. (C) Histogram of the overlap coefficients between the 131 shared iModulons between COLOMBOS and the combined dataset. (D) Scatter plot of the iModulon gene weights for the putative HprR iModulon. Purple genes are in both the iModulon from COLOMBOS and the iModulon from the combined dataset iModulon. Red genes are only in the iModulon from COLOMBOS, and blue genes are only in the iModulon from the combined dataset. The dashed lines indicate iModulon thresholds, and the gray diagonal line is the 45-degree line. (E) Schematic representation of the genes near *hprR*. (F) Bar chart of the putative HprR iModulon activities from GEO dataset GSE35371. (G) Scatter plot of the iModulon gene weights for an uncharacterized iModulon. Colors are identical to panel (d). (H and I) Relative iModulon activities of the iModulon from panel (G) from GEO datasets GSE21839 and GSE55365, respectively. Each dataset is centered to its own reference condition, so relative activities cannot be compared across bar charts. (J) Scatter plot of the iModulon gene weights for the antibiotic-responsive uncharacterized iModulon. (K and L and M) Bar chart of the antibiotic-responsive iModulon activities from GEO datasets GSE31140, GSE37026, and GSE10158.

though there is insufficient evidence to identify which regulator controls this iModulon, it is clear from the iModulon activities that these genes warrant further investigation.

In this section, we showed that ICA is scalable to thousands of expression profiles. Although the COLOMBOS dataset contains multiple E. coli strains, the overall iModulon structure was consistent with the iModulon structure identified from our combined microarray and RNA-seq compendium. In addition, we highlighted three iModulons with clear co-regulation evidence that are activated by hydrogen peroxide, anaerobic respiration, and antimicrobial agents, respectively. The myriad of uncharacterized genes in these iModulons are promising targets for in-depth functional characterization.

## Discussion

Matrix decomposition is a powerful approach to extract knowledge from large transcriptomics datasets. In particular, we have shown that ICA identifies highly similar structures between dissimilar datasets for the model bacteria *E. coli*. In addition, a combined compendium produced many identical iModulons as the individual datasets and could distinguish further signals that could not be identified in the separate datasets. The iModulons derived from the compendium can be applied to interpret new datasets, accelerating discovery and providing a standard framework that could be used to investigate any transcriptional regulator.

Throughout this study, we observed various properties of the iModulon decompositions: (1) iModulons co-occurring in multiple datasets tend to represent the effects of transcriptional regulators, (2) iModulon detection from a data set is dependent on the experimental conditions used to generate it, (3) ICA can be applied to cross-platform transcriptomic compendia without the need for normalization, (4) integration of data sets reveals iModulons not found in individual data sets, and (5) iModulons found in multiple independent datasets represent targets for regulon discovery.

In all five transcriptomic *E. coli* datasets, most iModulons could be characterized as representing the effects of a transcriptional regulator or a gene knock-out (i.e., Regulatory or Genomic). However, when extending to less-characterized organisms, it could be difficult to determine whether the remaining iModulons are technical artifacts or contain true biological insight, as demonstrated by the Central Dogma iModulon. This example demonstrates that if an iModulon is identified in multiple datasets, or if an iModulon persists upon addition of new datasets, then it could represent a true transcriptional signal.

Another important note is that the iModulons extracted from each dataset are sensitive to the experimental conditions that are represented in the dataset; ICA cannot extract an iModulon for a transcription factor whose activity never changes. Additionally, the CysB iModulons demonstrated that iModulons may represent the effects of multiple regulators, when the activities of the regulators are highly correlated across the measured conditions. However, adding new data or increasing the dimensionality of the decomposition can decouple the regulators, splitting the iModulon into its biological parts [24,51].

In contrast to the condition-invariant iModulon structure, iModulon activities represent the condition-dependent dynamics of expression profiles. In this study, we do not apply any normalization techniques to the data and clearly observe batch effects in the activity matrix of the combined compendium. Further work comparing identical experimental conditions from separate protocols is required to enable iModulon activity comparisons across datasets.

We have shown that the *E. coli* transcriptome contains a conserved, underlying structure that is found across multiple independent datasets. Previously, we also showed that this structure also exists across strains within a species [24]. This powerful observation enables unprecedented re-analyses of thousands of previously published datasets for a wide range of microbial organisms and demonstrates how data science can unlock hidden potential in complex biological datasets.

## Methods

### Rna-seq and microarray data processing

The full, log-transformed transcripts-per-million (log-TPM) for the RNAseq-1 dataset [24] was obtained from https://github.com/SBRG/precise-db. Raw data comprising the RNAseq-2 compendium were obtained from NCBI SRA under the bioproject accession number PRJNA379428 [34]. Raw data was processed using a similar process as described in Sastry et al. [24]. Raw sequencing reads were mapped to the reference genome (NC_000913.3) using bowtie (v1.1.2)[62] with the following options "-X 1000 -n 2–3 3". Transcript abundance was quantified using summarizeOverlaps from the R GenomicAlignments package (v1.18.0), with the following options "mode = "IntersectionStrict", singleEnd = FALSE, ignore.strand = FALSE, preprocess.reads = invertStrand"[63]. Transcripts per million (TPM) were calculated by DESeq2 (v1.22.1)[64]. The final expression dataset was log-transformed $\log_2$ (TPM + 1) before analysis, referred to as log-TPM.

CEL files were obtained for the three microarray datasets from NCBI GEO [1] (see S1 Dataset for accession numbers). In order to build large enough datasets for analysis, datasets MA-2 and MA-3 included all public data available from our research group that used the same expression platform. Each microarray dataset was normalized using robust multichip average (RMA) with default arguments from the R package *affy [65]*.

As the probes in microarrays may vary across platforms, only genes that were measured in all 5 datasets (3880 genes) were included in the analysis. Low quality expression profiles that clustered separately from the rest of the dataset were removed from the MA-1 and MA-2 datasets (See S1 Dataset). Each dataset was individually centered by subtracting the average expression profile across the replicates of a dataset-specific reference condition (see S1 Dataset). To create the combined compendium, the centered datasets were concatenated, without any additional normalization.

Only genes measured in all five datasets were retained, resulting in 3880 genes per dataset. Gene names, b-numbers, operons, and descriptions were obtained from Ecocyc [66]. Clusters of orthologous groups (COG) annotations were obtained from eggNOG 4.5.1 [67]. Additional annotations were obtained from Gene Ontology [68]. The transcriptional regulatory network (TRN) for transcription factors, small RNAs, and sigma factors was obtained from RegulonDB v10.0 [69] and supplemented with newly identified regulons from recent publications [24,44]. Riboswitches, tRNA-mediated attenuation, and dksA binding sites were obtained from Ecocyc [66] and UTP/CTP-dependent attenuation and reiterative transcription were obtained from Turnbough and Switzer [70]. All annotations for the 3880 genes are reported in S8 Dataset.

The COLOMBOS dataset was downloaded from http://colombos.net/ and all reference conditions were removed. Microarray profiles with over 300 empty values were discarded, and genes with any empty values in the remaining profiles were excluded. Replicate profiles with a Pearson correlation below 0.2 were also discarded, resulting in 3,016 final expression profiles.

### Independent component analysis

ICA decomposes a matrix ($\mathbf{X}$) into two matrices: $\mathbf{S}$ contains the independent signals, or structure, of the dataset, and $\mathbf{A}$ contains the condition-dependent activities of the signals:

$$\mathbf{X} = \mathbf{S}.\mathbf{A} \tag{1}$$

ICA was applied to each individual dataset and the combined compendium, as described in Sastry et al.[24]. Briefly, we executed FastICA 100 times with random seeds and a convergence tolerance of $10^{-6}$ for microarray and RNA-seq data, and a convergence tolerance of $10^{-3}$ for

COLOMBOS data. We constrained the number of independent components (ICs) in each iteration to the number of components that reconstruct 99% of the variance as calculated by principal component analysis. For the COLOMBOS dataset, we used the number of components that reconstructed 95% of the variance to reduce computation time. The resulting ICs were clustered using DBSCAN to identify robust ICs, with parameters with epsilon of 0.1, and minimum cluster seed size of 50. This process was repeated 10 times, and only ICs that consistently occurred in all runs were kept.

As described in Sastry et al.[24], iModulons were extracted from ICs by iteratively removing genes with the largest absolute value and computing the D'agostino $K^2$ test statistic of the resulting distribution. Once the test statistic fell below a cutoff, we designated the removed genes as the "iModulon".

To identify this cutoff for each individual dataset, we performed a sensitivity analysis on the concordance between significant genes in each IC and all known regulons. First, we isolated the 20 genes from each IC with the highest absolute gene weightings. We then compared each gene set against all known regulons using the one-sided Fisher's Exact Test (FDR $< 10^{-5}$). For each component with at least one significant enrichment, we selected the regulator with the lowest p-value.

Next, we varied the D'Agostino $K^2$ test statistic from 200 through 1000 in increments of 50. Using the protocol defined above, iModulons were extracted from ICs at each test statistic value, and the F1-score (harmonic average between precision and recall) was computed between the significant genes and its linked regulator. The test statistic with the maximum F1-score was used as the test statistic cutoff for the respective dataset.

## Characterizing iModulons

We compared the set of significant genes in each iModulon to each regulon (defined as the set of genes regulated by any given regulator) using the one-sided Fisher's Exact Test (FDR $< 10^{-5}$). We then compared the significant genes in each iModulon to the genes in each Gene Ontology (GO) term using the one-sided Fisher's Exact Test (FDR $< .01$). We added prophage information from Ecocyc to our GO database to capture iModulons representing prophages. In general, the final annotation was selected by the enrichment with the lowest q-value. Some iModulon annotations were manually curated, as denoted in S5 Dataset. Genomic iModulons were also manually curated by (1) comparing iModulon genes to known genetic alterations (e.g., knock-outs or overexpression), and (2) validating that the iModulon activities were affected in the appropriate direction for the corresponding strain.

## Comparing iModulon structures

To compare the complete structure of the transcriptomic datasets, we constructed the reciprocal best hit (RBH) graphs using the full IC gene weightings, rather than just the iModulon genes. We generated the RBH graph as described in Cantini et al. 2019 [33], using the following distance metric to compare ICs:

$$d_{x,y} = 1 - |\rho_{x,y}|$$

where $\rho_{x,y}$ is the Pearson correlation between components $x$ and $y$.

The Pearson correlation between two iModulons serves as a lower bound of the Szymkiewicz-Simpson overlap coefficient, which measures the level of overlap between two differently-sized sets (S8 Fig). We pruned all RBHs to remove links between ICs with similarities below 0.3, since the highest similarity between iModulons from the same dataset was 0.27. The full graph is shown in S1 Fig. The RBH graph was plotted using GraphViz [71].

### Linear regression of iModulons

Regression of the MA-3 CysB iModulon was performed using the LinearRegression function from Scikit-learn [72]. The ten ICs from the RNAseq-1 dataset with the highest absolute IC gene correlations with each of the CysB iModulon (See Table S1).

### Inference of iModulon activities

Raw reads for the ppGpp-RNAP dataset were downloaded from NCBI SRA (PRJNA504613) and processed as described above into log-TPM expression values. Two experimental conditions were selected for comparison, both using the wild-type strain with active *relA*, at 0 and 5 minutes after IPTG induction. Log-TPM expression values were averaged across triplicates.

To infer iModulon activities and calculate the amount of variance that iModulons explain between the two datasets, we first centered the two averaged expression profiles, then computed the gene-wise difference. The change in iModulon activity was calculated by multiplying the expression difference ($\Delta$**X**) by the pseudo-inverse of the S matrix from the full compendium:

$$\Delta A = pinv(S) \cdot \Delta X$$

Where *pinv* is the pseudoinverse function.

### Explained variance

Explained variance between two conditions was calculated as follows:

$$Explained\ Variance_k = \frac{\sum(\Delta X)^2 - \sum(\Delta X - S_k \Delta A_k)^2}{\sum(\Delta X)^2}$$

Where *k* is the iModulon of interest.

## Supporting information

**S1 Fig. Relative standard deviation (STD) of iModulon activities between replicates for each iModulon** in (a) RNA-seq datasets, and (b) microarray datasets.
(TIF)

**S2 Fig. iModulons in MA-3 that are linear combinations of iModulons in RNAseq-1.** Top scatter plot shows the MA-3 iModulon gene weights compared against the best hit iModulon in the RNAseq-1 dataset. Bottom scatter plot shows the linear combination of RNAseq-1 iModulons listed on the x-axis label.
(TIF)

**S3 Fig. Reciprocal best hit (RBH) graph containing all edges from the five datasets.** Edges with an IC Gene Weighting similarity score below 0.3 were pruned from Fig 2.
(TIF)

**S4 Fig. Characteristics of the only Genomic iModulon found in more than one dataset.** (a) Activities of the iModulon in the RNAseq-1 dataset separate E. coli strains evolved for growth on the non-native carbon source D-lyxose from the other strains in the compendium [1]. (b) Activities of the uncharacterized iModulon in MA-1 that was linked to the iModulon described in panel (a). Seven strains were evolved in parallel for growth on D-lactate [38], but only two endpoint strains (named Lac2 and Lac3) exhibited high iModulon activities. These strains were not re-sequenced, so the adaptive mutations could not be confirmed. (c) Scatter

plot showing the IC gene weightings corresponding to the iModulons described in panel (a) (in black) and panel (b) (in red). Thresholds determining iModulon composition are indicated by dashed lines. The genomic duplication from the D-lyxose-evolved strains is highlighted in gray, indicating that all strains with high activities likely acquired an identical duplication along their evolutionary trajectory. Two transcription units outside of the duplicated region were captured in both iModulons, the *galETK* transcription unit, and the *galP* gene. These genes are responsible for D-galactose catabolism.
(TIF)

**S5 Fig. Investigation of a complex iModulon cluster.** Each dataset contains a different set of conditions, which can activate different groups of respiration-related genes leading to different co-expression patterns between datasets. (a) The complex iModulon cluster shows that two iModulons from the RNAseq-1 dataset are indirectly connected. (b) Scatterplots of the IC gene weights for the iModulons highlighted in (a). The Pearson R correlation of the IC gene weights is shown below. The two RNAseq-1 iModulons show no correlation, but still contain a few genes in common, indicating that the expression of these shared genes are controlled by two distinct underlying sources.
(TIF)

**S6 Fig. Reciprocal best hit (RBH) graph of Principal Components from all 5 transcriptomic datasets.** Edges with a gene weighting similarity score below 0.3 were pruned from this figure. Few iModulons have a reciprocal best hit, and no conserved clusters exist when analyzing Principal Components.
(TIF)

**S7 Fig. Effects of data integration on iModulon structure and activity.** (a) Histogram of the percent of total expression explained by each independent component in the individual datasets. Components that are maintained (i.e., identify an RBH hit with the full compendium decomposition) are colored in pink, whereas components that are lost upon data integration are colored in blue. (b) Pie chart illustrating the classes of the new iModulons extracted from the combined datasets. (c) Histogram of the IC gene coefficients of a Single Gene component. Dashed lines indicate the iModulon threshold. (d) Schematic illustration of precision, recall, and F1-score. (e) Boxplots of the F1-scores between iModulons and their associated regulators for Regulatory iModulons. Only Regulatory iModulons in the individual dataset that found an RBH in the full compendium are shown in the left boxplot, and the RBH of these iModulons in the full dataset are shown in the right boxplot. (f) Schematic illustration of the various processes encoded by the genes in the Central Dogma iModulon. (g) Time-course treatment of DNA-damage inducing norfloxacin activates an iModulon that shows reduced activity when RNAP is bound by ppGpp. (h) Histogram of absolute Spearman correlations between iModulon activities in components that are RBHs in the full dataset compared to the individual datasets.
(TIF)

**S8 Fig. Comparison of Szymkiewicz-Simpson overlap coefficient and Pearson R correlation coefficient for linked iModulons.**
(TIF)

**S1 Table. Reciprocal best hits between the iModulons from the five transcriptomic datasets.**
(XLSX)

**S2 Table. iModulons from RNAseq-1 that combine to form the MA-3 CysB iModulon.**
(XLSX)

**S3 Table. Reciprocal best hits between the iModulons from the combined transcriptomic compendium and the iModulons in the individual datasets.**
(XLSX)

**S1 Dataset. Experimental conditions and metadata for the five expression datasets.**
(XLSX)

**S2 Dataset. Expression levels for RNA-seq datasets.**
(XLSX)

**S3 Dataset. Expression levels for microarray datasets.**
(XLSX)

**S4 Dataset. IC gene weightings for individual RNA-seq and microarray decompositions.**
(XLSX)

**S5 Dataset. IC gene weightings for combined compendium decomposition.**
(XLSX)

**S6 Dataset. iModulon activities.**
(XLSX)

**S7 Dataset. iModulon characterization and constituent genes.**
(XLSX)

**S8 Dataset. Annotations, ontologies, and regulon memberships for all 3880 genes in the datasets.**
(XLSX)

## Acknowledgments

The authors would like to thank Dan Zielinski, Eivind Almaas and Yara Seif, and CJ Norsigian for informative discussions. This research used resources of the National Energy Research Scientific Computing Center, a DOE Office of Science User Facility supported by the Office of Science of the U.S. Department of Energy under Contract No. DE-AC02-05CH11231.

## Author Contributions

**Conceptualization:** Anand V. Sastry, Bernhard O. Palsson.

**Data curation:** Anand V. Sastry, Alyssa Hu, David Heckmann.

**Funding acquisition:** Bernhard O. Palsson.

**Investigation:** Anand V. Sastry, Alyssa Hu, David Heckmann, Saugat Poudel, Erol Kavvas.

**Methodology:** Anand V. Sastry.

**Resources:** Bernhard O. Palsson.

**Supervision:** Bernhard O. Palsson.

**Visualization:** Anand V. Sastry, Saugat Poudel, Erol Kavvas.

**Writing – original draft:** Anand V. Sastry, Bernhard O. Palsson.

**Writing – review & editing:** Anand V. Sastry, David Heckmann, Saugat Poudel, Erol Kavvas, Bernhard O. Palsson.

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
