## [Decision Letter · Decision Letter 0]

20 Oct 2020

Dear Prof. Palsson,

Thank you very much for submitting your manuscript "Matrix factorization recovers consistent regulatory signals from disparate datasets" for consideration at PLOS Computational Biology.

Our apologies for the delay in decision. We have had difficulties in securing reviews on time - likely due to the current pandemic situation that has placed much burden on the academic community. We thank you for your patience and understanding.

Your manuscript was reviewed by members of the editorial board and two independent reviewers. In light of the reviews (see my own comments next and the reviewers' comments below this email), we would like to invite the resubmission of a significantly-revised version that takes into account the reviewers' comments.

We cannot make any decision about publication until we have seen the revised manuscript and your response to the reviewers' comments. Your revised manuscript is also likely to be sent to reviewers for further evaluation.

Sincerely,

Kiran Raosaheb Patil, Ph.D.

Associate Editor

PLOS Computational Biology

Weixiong Zhang

Deputy Editor

PLOS Computational Biology

Editor comments:

Major comments:

1. There is no analysis/discussion regarding the differences that can (and most likely will) arise from grouping different strains together. Strains resulting from ALE can harbour substantial differences in regulatory program, and this effect can be much greater in natural variants (even the genome size can differ by >10%). I recommend to discuss this, whenever possible with supporting additional analysis.

2. Given that future application for the method is going to be either much larger datasets or other species (as authors point out), I think the manuscript would much benefit by expanding the scope in this direction. This could be done, e.g., by analysing datasets for additional species of health/industrial importance, and/or by including cross-species comparison, and/or by increasing the number of datasets. This would also help addressing the questions of novelty raised by the other reviewer.

Minor comments:

1. Fig 2: X-axis labels unclear. R would be better indicated inside the plot and not as a x-axis label.

2. Table 1: Please consider removing the column ‘Research Group’. This is redundant with the next column (reference) and also unnecessarily gives an impression of subjectivity.

Reviewer's Responses to Questions

**Comments to the Authors:**

Reviewer #1: The paper by Sastry et al presents some results based on the use of ICA to recover regulatory signals from different datasets. The topic is a relevant one, and overall the study presented is, in my opinion, technically sound and presents some technical details useful for the community. Still, I believe that the results of the paper do not represent a major breakthrough to warrant its publication in PLOS CB.

Indeed, the main contribution of the paper over previous work is, as stated in the last paragraph of the Introduction, to show that the method provided in previous work based on ICA is capable of retrieving consistent regulatory modules from different transcriptomics and proteomics datasets for E. coli. Although some results point to consistent modules, I believe that the work shown here is still preliminary, showing somewhat marginal contributions over previous work, and some of the claims need more support. Some reasons for this statement are listed below:

- the analysis of the consistence of the iModulons obtained in based on Pearson correlation of the coefficients of the ICA (a metric of linear correlation while ICA can capture nonlinear relationships; maybe mutual information would be an option);

- a match of different modules in different datasets is obtained based on reciprocal best hit; no analysis is provided on the number of shared genes. Also, RBH is done pairwise and thus the 45% of linked iModulons is an interesting result, but not of a level that justifies some of the produced claims. The number of modules consistent in all 5 datasets (or even 3-4) seems to be low (even the example presented of CysB only holds for 4 datasets), although this is not discussed.

- the RBH by itself does not represent a high consistency between the connected modules, only that they are the best candidates to be connected.

- it is not obvious why almost half of the modules retrieved from microarrays datasets cannot be characterised.

- the results shown for the proteomics datasets seem to be very inconclusive and could be better discussed; they do not seem to justify the initial claim (in the abstract: "echoes of this structure remain in the proteome, accelerating biological discovery through multi-omics analysis"); also, the final sentence of that section is rather confusing.

- comparisons with known regulators and regulated genes are only done via statistical tests/ enrichment and it woud be interesting to compare possible composition of extracted modules as sets of genes. It would thus be interesting to see if this method could capture regulatory modules that could be used for predictive tasks.

- the code in github is only partial; the full reproducibility of the results, although possible in theory is made harder since the scripts run to generate the results are not made available.

Also, some minor aspects:

- in the Introduction (3rd paragraph; sentence starting with "Previously, ..." it is not clear that when describing previous work that it comes from ref 36, not cited there; given the importance of this previous work, it should be more clear.

Finally, I note that since the study works over the same organism, it is "known" that the regulatory modules should be shared (while we may not know all of them) and are in fact underlying the generation of all these datasets. So, it would be surprising to find that the modules retrieved from these datasets would not contain some similarities. So, in a way, what would be surprising would be the lack of this consistency.

**Have all data underlying the figures and results presented in the manuscript been provided?**

Reviewer #1: Yes

PLOS authors have the option to publish the peer review history of their article (what does this mean?). If published, this will include your full peer review and any attached files.

Reviewer #1: No

Reviewer#2:  Summary of the paper:

The authors have used five independently produced datasets of RNAseq and microarray, and performed an ICA analysis on them to find underlying consistent patterns in latent space. Each independent component consists of weights of genes, and the few genes with a significant weight are grouped as iModulons. The iModulons were then characterized as regulatory, functional, genomic, or uncharacterized. Some of these iModulons were found in all datasets, and many were found in at least two, indicating that this technique can find relevant features of genetic data.

Comments:

Major

My main comment is novelty, the authors use same technique/same philosophy published recently (ref 26) and similar research design, i.e. multiple cancer datasets comparison, ref 35, so message Matrix factorization recovers consistent regulatory signals from disparate datasets is not new. I would recommend to authors focusing on biological story, otherwise from low-rank matrices is pretty clear that they are the similar between biological experiments, especially when dealing with linear techniques such as ICA. Also although authors did analysis on log-transformed data, I would like to see it this analysis holds true on standardized data (mean-cantered and scaled to unit variance), otherwise ICA is sensitive to scale of data and genes that are highly expressed in one dataset are also same genes that are highly expressed in another. Otherwise, technically paper sounds well and nicely written with nice graphics.

Minor

I recommend to change short tittle “iModulons are conserved across disparate datasets” - iModulons is not something established, neither informative.
---

## [Decision Letter · Decision Letter 1]

18 Dec 2020

Dear Prof. Palsson,

We are pleased to inform you that your manuscript 'Independent component analysis recovers consistent regulatory signals from disparate datasets' has been provisionally accepted for publication in PLOS Computational Biology.

Best regards,

Kiran Raosaheb Patil, Ph.D.

Associate Editor

PLOS Computational Biology

Weixiong Zhang

Deputy Editor

PLOS Computational Biology

Reviewer's Responses to Questions

**Comments to the Authors:**

Reviewer #2: The revised manuscript carefully takes into account the comments I raised previously. It appears to be now appropriate for publication in Plos Computational Biology

**Have all data underlying the figures and results presented in the manuscript been provided?**

Reviewer #2: Yes

PLOS authors have the option to publish the peer review history of their article (what does this mean?). If published, this will include your full peer review and any attached files.

Reviewer #2: No

---

## [Editor Report · Acceptance letter]

26 Jan 2021

PCOMPBIOL-D-20-00992R1 

Independent component analysis recovers consistent regulatory signals from disparate datasets

Dear Dr Palsson,

I am pleased to inform you that your manuscript has been formally accepted for publication in PLOS Computational Biology. Your manuscript is now with our production department and you will be notified of the publication date in due course.

With kind regards,

Alice Ellingham
